# Advanced Neural Interface toward Bioelectronic Medicine Enabled by Micro-Patterned Shape Memory Polymer

**DOI:** 10.3390/mi12060720

**Published:** 2021-06-19

**Authors:** Youngjun Cho, Heejae Shin, Jaeu Park, Sanghoon Lee

**Affiliations:** Robotics Engineering, Daegu Gyeongbuk Institute of Science and Technology (DGIST), Daegu 42899, Korea; cyj003450@dgist.ac.kr (Y.C.); volt11@dgist.ac.kr (H.S.); bnm5641@dgist.ac.kr (J.P.)

**Keywords:** neural interface, shape memory polymer, neuromodulation, neural recoding, neural stimulation, bioelectronic medicine

## Abstract

Recently, methods for the treatment of chronic diseases and disorders through the modulation of peripheral and autonomic nerves have been proposed. To investigate various treatment methods and results, experiments are being conducted on animals such as rabbits and rat. However the diameter of the targeted nerves is small (several hundred μm) and it is difficult to modulate small nerves. Therefore, a neural interface that is stable, easy to implant into small nerves, and is biocompatible is required. Here, to develop an advanced neural interface, a thiol-ene/acrylate-based shape memory polymer (SMP) was fabricated with a double clip design. This micro-patterned design is able to be implanted on a small branch of the sciatic nerve, as well as the parasympathetic pelvic nerve, using the shape memory effect (SME) near body temperature. Additionally, the IrO_2_ coated neural interface was implanted on the common peroneal nerve in order to perform electrical stimulation and electroneurography (ENG) recording. The results demonstrate that the proposed neural interface can be used for the modulation of the peripheral nerve, including the autonomic nerve, towards bioelectronic medicine.

## 1. Introduction

As neural interfacing technology advances, many treatments for chronic diseases and disorders have been proposed through neural stimulation of the peripheral nervous system (PNS), including the autonomic nervous system (ANS). For instance, the possibility of treating chronic diseases such as Crohn’s disease, rheumatoid arthritis, obesity, and diabetes through the modulation of the vagus nerve (VN) has been reported [1,2,3,4]. In addition, a therapeutic alternative method for the neurogenic bladder has been suggested through the modulation of the bladder nerves including the parasympathetic nervous system (PSNS) [5,6]. As ANS is deeply involved in both human survival and health, safe and reliable interaction with these nerves is one of core technologies of this field. However, it remains a current challenge. Advanced interfacing technologies include: (i) precise control of specific functions through neural stimulation; (ii) quick and mechanically secure implantation in the presence of physiological motion such as respiratory and cardiovascular movements; and (iii) considerable compliance and flexibility from the neural interface, since nerves are highly compliant and associated with moving organs [5]. This requires careful consideration of material properties, electrode design, electrode configuration, as well as a method of implantation. Furthermore, all these considerations should apply to very small scale of nerves that are vulnerable to damage.

The neural interface can be divided into invasive (intra-neural) and non-invasive (extra-neural) types [7,8] and, due to the uniqueness of autonomic nerves, non-invasive extra-neural interfaces are preferred. To design an advanced extra-neural interface for this application, careful considerations of interface design are required such as stiffness, thickness, shape of the interface, and electrode configuration in order to function properly and reliably. Since the extra-neural interface surrounds the nerve, the neural tissue and the neural interface contact with a large area. Accordingly, the stiffness of the substrate is important, and the mechanical mismatch between them, which directly causes foreign body reactions, becomes a more important factor. The scar tissue which formed by this immune response interferes with the chronic implantation of the neural interface [9]. Since the Young‘s modulus of neural tissue is very soft (10 kPa), the neural interface was fabricated by using soft materials such as silicone elastomer [10]. However, the difficulty of micro-patterning of the soft material leads to the thicker neural interfaces (>100 μm) and simple electrode configuration, which results in design limitations. For instance, a thicker substrate makes implantation into small nerves difficult due to the higher bending stiffness of the device [11]. Also, the thickness (200~600 μm) of a neural interface can worsen the fibrotic response [12]. As the diameter of the target nerve is typically small, a design that can be implanted into exceedingly small nerves such as the pelvic nerve (PN; 100~200 μm) of rodents is required. To this end, various types of neural interfaces have been fabricated by micropatterning materials such as polyimide (PI). However, although the PI based neural interface is thin enough (10~20 μm), it is very stiff (>2 GPa) compared to the nerve and there is still a mechanical mismatch [11,13].

Biocompatibility for the used thiol-ene/acrylated-based shape memory polymer (SMP) was demonstrated by in vitro cytotoxicity testing, and showed suitable results as a substrate for neural interfaces [14]. SMPs are polymers that can recover their original programmed shape when exposed to external stimuli such as temperature and electricity. Additionally, SMP’s have a softening effect, based on glass transition temperature (T_g_). Furthermore, the temperature at which the shape memory effect of SMP occurs is defined around the T_g_ [15]. These characteristics of SMPs enable micropatterning with sufficient Young’s modulus (1.5 GPa) at room temperature and have the effect of softening when implanted in the body. In addition, the shape memory effect (SME) can be used as a method of implantation by transforming the interface design into a 3D structure for insertion into a nerve, and then can recover its shape at body temperature for conformal contact and fixation. Recently, a neural interface capable of self-implanting into nerves through the SME of SMPs programmed with T_g_ near body temperature was reported [16]. Reported neural interfaces have added 100 μm thick SMP substrates for SMEs to PI-based neural interface. However, this thickness is difficult to implant into a small nerve such as an autonomic nerve. Another neural interface using SMP was also reported, which had a thickness of only several tens of micrometers and was implanted into autonomic nerves such as PN and VN [12]. In addition, electrodes of various sizes were patterned through micropatterning at a thin thickness of SMP, and the immune response of nerves was reduced through the softening effect of SMP. However, since the T_g_ of the used SMP was much higher than body temperature, it is difficult to implement functional implantation through the SME during the implantation.

In this paper, using the characteristics of SMPs, a double clip neural interface (DCNI) that is thin (40 μm), soft (400 MPa), easy to implant due to self-clipping, and stably fixed to the nerve with a double clipping structure was developed. An SMP with T_g_ near body temperature was used as a substrate. DCNI micropatterned by SMP was fabricated through standard photolithography processes. Also, iridium oxide (IrO_2_) was coated to improve electrochemical performances. The DCNI was designed with a double clipping design for rigid fixation and was designed to be reshaped with a 3D hook structure. In addition, it was designed to clip by itself after implanting into the nerve with a 3D hook structure using SME occurring near body temperature. This DCNI’s structural and functional design allows for easy implantation. To demonstrate the implantation performance of the DCNI, the DCNI was implanted into the sciatic nerve (SN) branch and PN of the rodents. In addition, in vivo experiments of neural stimulation and neural recording using the DCNI implanted on the common peroneal nerve (CPN) were investigated.

## 2. Materials and Methods

### 2.1. Materials and SMP Synthesis

1,3,5-Triallyl-1,3,5-triazine-2,4,6(1H,3H,5H)-trione (TATATO) (Sigma-Aldrich, Seoul, Korea), trimethylolpropanetris(3-mercaptopropionate) (TMTMP) (Sigma-Aldrich, Seoul, Korea), tricyclodecanedimethanoldiacrylate (TCMDA) (Sigma-Aldrich, Seoul, Korea), and 2,2-dimethoxy-2-phenyl-acetophenone (DMPA) (Sigma-Aldrich, Seoul, Korea) were used for SMP synthesis. The synthesis procedure was conducted in a class 1000 clean room. The SMP was synthesized by mixing stoichiometric quantities of the TATATO and TMTMP with 31 mol% TCMDA and 0.1 wt% of DMPA as a photoinitiator. The solution was mixed through a paste mixer (AR-100, THINKY, Laguna Hills, CA, USA) to obtain a uniform solution. The solution was spin coated at 800 rpm for 25 s (20 μm) on Al coated Si wafer by a spin coater (SPIN-1200T, MIDAS, Seongnam, Korea) and cured in 365 nm UV for 1 h and 120 °C vacuum oven for 24 h.

### 2.2. Fabrication Process of the DCNI

The 1st SMP layer was spin coated and cured at the thickness of 20 μm on the Al coated Si wafer. Al was used for a sacrificial layer. An adheshion layer (Cr) of 50 nm and a conducting layer (Au) of 200 nm were deposited by a RF sputter, and were patterned by standard photolithography process with photoresist (DNR-L300-40, Dongin-Semichem, Seoul, Korea). The 2nd SMP layer was also coated and cured at 20 μm. A hard mask (Cr) of 200 nm was patterned using the same method as the conducting layer. Then, the SMP was etched by a reactive ion etching (RIE) with O_2_ plasma. The sacrificial layer (Al) and the hard mask (Cr) were removed using HCL 2M solution and a chrome etchant (Sigma-Aldrich, Seoul, Korea), respectively (Figure 1a). The fabricated DCNI is shown at Figure 1b.

### 2.3. Dynamic Mechanical Analysis (DMA)

The storage modulus (E′) and tan δ were measured in order to evaluate the mechanical properties of the SMP using a dynamic mechanical analyzer (DMA Q800, TA Instruments, New Castle, DE, USA). The measured sample was prepared by a laser, cutting a thin SMP film with a thickness of 20 μm into a rectangular of size 40 mm × 6 mm. Measurements were conducted at a 0.2 N preload force at 1 Hz with 0.275% strain and the temperature changes were −10 °C to 80 °C with 2 °C min^−1^ heating rate.

### 2.4. Iridium Oxide (IrO_2_) Coating

To make the solution for IrO_2_ coating, 300 mg of the iridium chloride was added to 200 mL of the DI water and stirred for 15 min. 1000 mg of the oxalic acid powder was added to the solution and stirred for 10 min. Potassium carbonate was used for controlling the pH of the solution to 10.5. The prepared solution was rested at room temperature for 2 days, during which it became a violet color. IrO_2_ was electrodeposited through a cyclic voltammetry (CV) measuring instrument using a three-electrode configuration. Only the counter electrode and working electrode were used. The reference electrode was not used. A platinum mesh electrode was used as a counter electrode and a neural interface was used as a working electrode. A voltage of 0 V to 0.76 V was applied at a scan rate of 50 mV/s, and was swept 50 times.

### 2.5. Characterization of the Nerual Interface

To investigate the characteristics of the DCNI electrochemical impedance spectroscopy (EIS) was conducted using multichannel potentiostat (Ivium-n-Stat, IVIUM Technology). The Three-electrode configuration was applied where a sliver/silver chloride (Ag/AgCl) electrode was used for the reference electrode, and a platinum (Pt) electrode was used for the counter electrode. Phosphate buffered saline (PBS) was used as the medium and a scan rate of 50 mV/s and frequencies from 1 Hz to 100 kHz were applied.

### 2.6. Animals

A 12 weeks old Sprague Dawley (SD) rat was used for the experiment. For recording and stimulation in an acute level, an electromyography (EMG) wire interface, DCNI, and stainless steel hook electrode were implanted into the tibialis anterior (TA) muscle, CPN, and SN, respectively. The DCNI was implanted into the PN to confirm that easy implantation is possible.

### 2.7. Ethics Statement

Animal cares and use protocol was reviewed and approved by the Institutional Animal Care and Use Committee at the Daegu Gyeonbuk Institute of Science and Technology (Approval No. DGIST-IACUC-21012702-0001).

### 2.8. Surgical Procedures

To induce anesthesia in the animals, 20 mg/kg of ketamine was injected intraperitoneally. Animal was induced by inhalation anesthesia with vaporized isoflurane (1%) in a constant oxygen flux. For the SN, the hind limb was incised 4 cm along the femur and the femoral muscles were separated. The connective tissue and fat around the nerves were cleared and the space was reserved for the implantation of the neural interface. The DCNI was implanted into the CPN. For the PN, a 3 cm incision was made vertically at 1 cm on the right side of the pubis. The bladder and surrounding vein were anatomically referenced to find the PN. The connective tissue and vein around the nerves were cleared and the DCNI was implanted into the PN. After the experiment, the animals were euthanized in vaporized isoflurane (5%) in a constant oxygen flux.

### 2.9. Signal Acquisition and Processing

The RHD 2216 (Intan Technologies, LA, USA) was used for EMG acquisition at a sampling rate of 30 kHz. A 60 Hz notch filter and a band-pass filter with cutoff frequencies of 0.1 Hz and 2.5 kHz were applied in signal recording. The RHD 2132 (Intan Technologies, Los Angeles, CA, USA) was used for compound neural action potential (CNAP) acquisition at sampling rate of 30 kHz. A notch filter and band-pass filter were applied in the same way as the RHD 2216. The acquired signal was signal-processed using MATLAB 2020a software. Sixty evoked EMGs from a TA muscle and CNAPs from a CPN in every second were recorded and averaged to reduce noise.

## 3. Results

### 3.1. In Vitro Characterization of the Nerual Interface

#### 3.1.1. Mechanical Characterization

An analysis of the mechanical property of the SMP substrate (μm), DMA testing was conducted. Figure 2a shows the storage modulus of the SMP. The storage modulus of the SMP was 1.4 GPa in room temperature (22 °C). This level of stiffness enabled complex micropatterning by photolithography processes. The SMP began to soften rapidly from 30 °C due to the properties of polymer, and its Young’s modulus at body temperature (37 °C) was 400 MPa. The glass transition temperature was 44 °C (Figure 2b).

#### 3.1.2. Electrochemical Characterization

To analyze the performance of the DCNI, an EIS test was conducted. The impedance and CSC were compared with only the Au deposited neural interface and IrO_2_ coated neural interface. To measure the impedance of the neural interface, frequencies from 1 Hz to 100 kHz were applied. The impedance of the non-coated neural interface and IrO_2_ coated neural interface at 1 kHz were 30.74 kohm and 2.114 kohm, respectively (Figure 3a). To measure the stimulation performance of the neural interface, CSC was measured by CV. −0.6 V to 0.8 V was applied, and the scan rate was 50 mV/s. The CSC of the non-coated neural interface and the IrO_2_ coated neural interface were 0.19 mC/cm^2^ and 14.59 mC/cm^2^, respectively (Figure 3b). The IrO2 coating reduced the impedance 14.54 times and increase the CSC 76.77 times.

#### 3.1.3. Stability Test

A stability test was performed in order to confirm that the IrO2 coating was stably electrodeposited. A continuous voltage was applied to the DCNI in PBS for 2 h. A biphasic triangular pulse with a maximum amplitude of 0.8 V and a minimum amplitude of −0.6 V was applied once per second in the same method as the CV measurement method with the potentiostat above. Impedance and CSC were measured at 30 min intervals to check the state of the electrodes, and the measured results were normalized (Figure 4a). After 2 h, the impedance of the DCNI decreased by 9.64% and CSC increased by 8.45%. The state of the coated electrode surface was indirectly confirmed by fitting a CV curve over time (Figure 4b). No significant changes were observed in the CV curve and size. No degradation of DCNI performance was observed with long term voltage application.

Another stability experiment was performed in order to determine the effect of the bending process performed upon the implantation of the DCNI. Impedance and CSC were measured in the initial state of the DCNI, in the bending state deformed into a hook shape through the bending process, and in the recover state where the shape was restored through the SME. All measured results were normalized. Impedance increased by 8.84% and CSC decreased by 6.47% by bending stress generated in the bending state. After recovery at 38 °C, the impedance was increased by 0.39% and the CSC was increased by 0.65% compared to the initial state (Figure 4c). Although the bending stress generated under the bending process affected the performance of DCNI, the performance change after recovery was very insignificant (less than 1%), confirming that there is no problem in stimulation and recording performance.

### 3.2. Desgin and Functional Implantation Experiment

#### 3.2.1. Design of the DCNI

The design of the DCNI was intended to complement the shortcomings of the existing clip interface [5]. In the case of the clip interface, it was designed under the inspiration of a paper clip. In the case of the existing clip interface, there was a disadvantage insofar as the nerve could not be fixed under the clip portion of the clip interface after the electrode was clipped to the nerve. Therefore, it moved along the nerve during chronic implantation. To solve this issue, a design that can double-clip the upper and lower parts of the nerve was devised and applied to the DCNI (Figure 5a). In the case of relatively stiff materials such as PI, the design has to be clipped twice to the nerve, so there is difficulty in implantation. However, in the case of the DCNI, it was designed to enable double clipping by itself after converting to a 3D structure using SMP, which enables easy implantation and stable fixation of electrodes.

#### 3.2.2. Functional Implantation Process

The fabricated DCNIs were implanted into the CPN (Figure 5b) and PN (Figure 5c) to confirm the functional effect. For easy implantation, a reshape procedure was conducted. To minimize the damage, the DCNI was heated on the 50 °C hot plate. And the shape of the DCNI was changed like a hook and fixed on the 0 °C. The hook-shaped DCNI was held on the nerve and 38 °C PBS was spread on the DCNI. Because of the shape memory effect, self-clipping of the DCNI occurred and the DCNI was fixed onto the nerve (Figure 5a).

### 3.3. Stimulation and Recording

Stimulation and recording were conducted on the SD rat on an acute level. A stimulation test was performed to confirm nerve stimulation using DCNI. The nerve was stimulated through the DCNI implanted into the CPN, and the muscle signals were recorded using a stainless steel wire implanted into the TA muscle (Figure 6a). The biphasic waveform of 100 μs pulse was applied at 1 Hz for 1 min, and the current amplitudes were changed from 50 μA, 100 μA, to 150 μA, respectively. Muscle twitching following stimulation was confirmed. The EMG signal and signal process method as mentioned in Section 2.9 was acquired, and the processed EMG signal is shown in Figure 6b. The peak-to-peak of the acquired EMG signal was 1120.4 μA at 50 μA, 1513.8 μA at 100 μA, and 2221.7 μA at 150 μA, confirming that the magnitude of the acquired EMG signal increased as the stimulation intensity increased (Figure 6c).

The recording test was conducted to confirm the recording ability of the DCNI. CNAPs were recorded through the DCNI on the CPN by stimulating hook electrode on the SN. Electrical stimulation was conducted by hook electrode and the stimulation parameter was the same as stimulation test. Leg twitching due to stimulation was confirmed. The electroneurogram (ENG) signal was acquired and the signal process method as mentioned in Section 2.9 was conducted. The processed ENG signal is shown in Figure 6d. The peak-to-peak of the acquired ENG signal was 1308.2 μA at 50 μA, 1498.1 μA at 100 μA, and 1544.9 μA at 150 μA. The results indicate that the stimulation parameters are already exceeding the threshold energy, and that the amplitudes of evoked compound neural action potentials (CNAPs) become saturated as the stimulation amplitude increases (Figure 6e). The overall results demonstrate that the DCNI fabricated by SMP shows good implantation performance using SME and can be used for neural recording and stimulation.

## 4. Discussion

We developed the DCNI using a thiol-ene/acrylate based SMP polymer that can be easily implanted into small nerves, such as a branch of the sciatic nerve and the parasympathetic pelvic nerve, using SME. This neural interface was fabricated by a photolithographic process, and the proposed process enables complex micropatterning of several micrometers. Through this process, SMP neural interfaces of various designs can be fabricated for various sizes of peripheral nerves including the autonomic nerves. The fabricated DCNI showed a Young‘s modulus of 400 MPa (at 37 °C), which has softer properties than polyimides conventionally used as neural interface substrates. In addition, since this polymer was much softer in wet conditions [12], softer material property than the 400 MPa (at 37 °C) is expected during long term implantation. Toward bioelectric medicine applications, the size of the target nerve to be modulated is gradually getting smaller. Small nerves are easy to damage, and the corresponding neural interface becomes thinner and smaller, such that even a small impact can cause fatal damage. Therefore, a neural interface that can be easily and safely implanted (even for small nerves) is required. In that respect, it is possible to change the DCNI to a 3D structure using the characteristics of SMP, and the recovery effect around T_g_ has the advantage of being able to easily implant neural interfaces, decreasing damage of the nerves and electrodes even in small nerves. To prove this, the DCNI was easily implanted into SN and PN through the proposed function implantation process.

Since the overall size of the electrodes is also decreased with the reduction of nerve sizes, improved conductive material with lower resistance and higher charge transfer performance than gold is required. To this end, IrO_2_ was electrodeposited to increase CSC 76.77 times and increase the stability of the DCNI during electrical stimulation. This also increases the recording performance because the impedance decreased by a factor of 14.54. To confirm these performances, the DCNI was implanted into the CPN, and both the muscle activation through electrical stimulation and the evoked compound action potential (CNAP) recording were successfully performed.

The DCNI not only satisfies various conditions such as biocompatibility, thin thickness, a lower Young’s modulus, and a simplification of the fabrication process using micropatterning with a photolithography process, but also enable easily implants with a 3D structure into small nerves using the characteristics of SMPs. These characteristics of the DCNI are promising as an advanced neural interface. In conclusion, the DCNI was developed for the modulation of PNS and ANS, and we expect that it can be used for neuromodulation for disease treatment in future.

## Figures and Tables

**Figure 1 micromachines-12-00720-f001:**
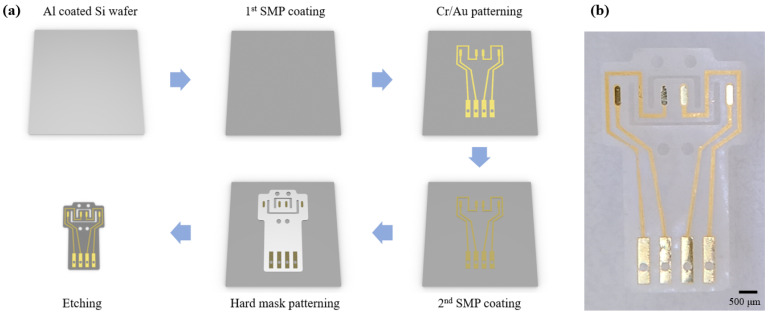
Fabrication process of the DCNI: (**a**) Schematic of the fabrication procedure; (**b**) Fabricated DCNI.

**Figure 2 micromachines-12-00720-f002:**
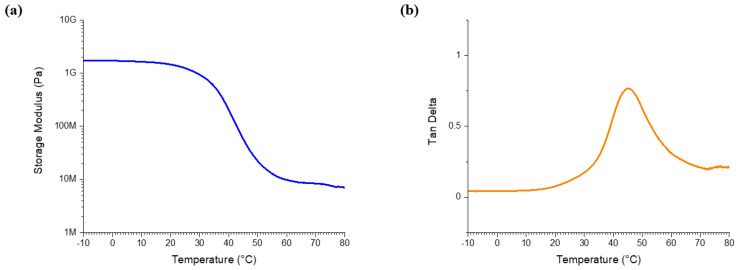
The characterization of shape memory polymer (SMP): (**a**) Storage modulus and (**b**) Loss factor (Tangent δ).

**Figure 3 micromachines-12-00720-f003:**
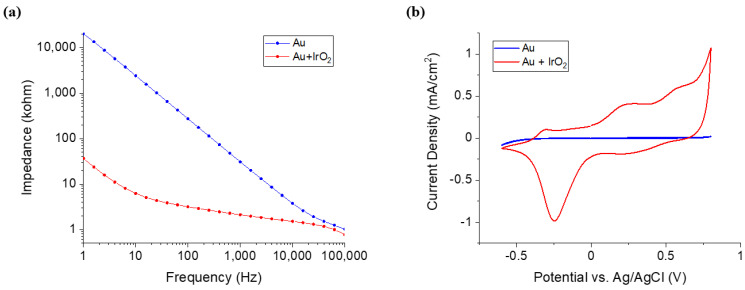
Characterization of DCNI (Au vs IrO_2_ coating): (**a**) Impedance measurement of the DCNI and (**b**) CV test of the DCNI.

**Figure 4 micromachines-12-00720-f004:**
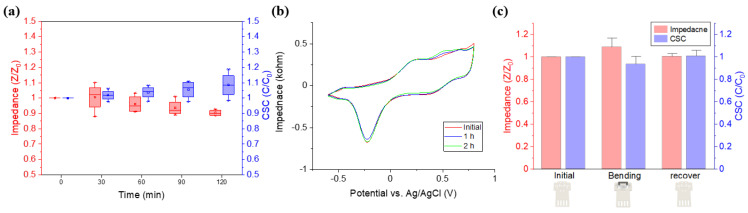
Stability test of the DCNI: (**a**) Impedance and CSC change according to volage application time (n = 4); (**b**) CV curve according to volage application time; and (**c**) Stability of the bending process (n = 4).

**Figure 5 micromachines-12-00720-f005:**
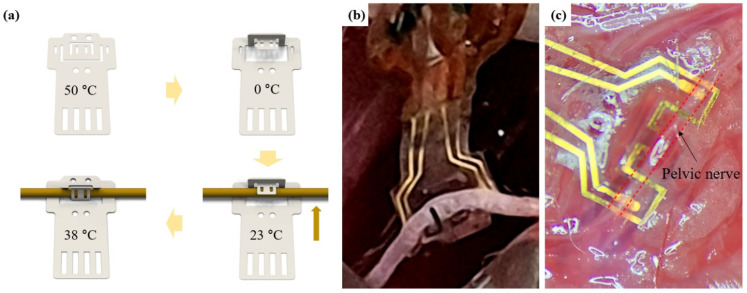
Functional implantation in bench up and in vivo: (**a**) Schematic of DCNI implantation process; (**b**) CPN implantation of the DCNI; and (**c**) PN implantation of the DCNI.

**Figure 6 micromachines-12-00720-f006:**
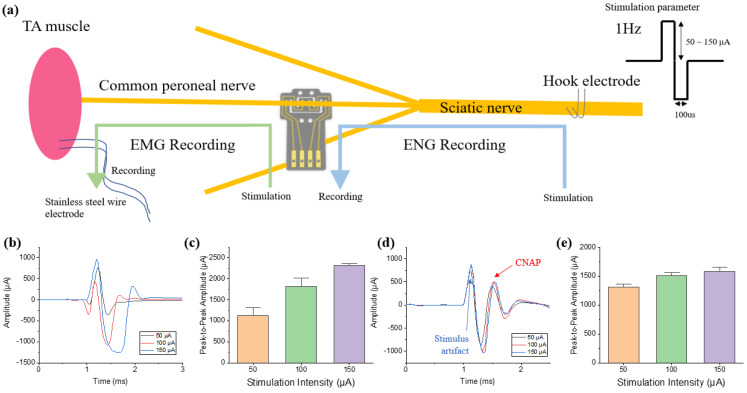
In vivo experiments of neural signal recording and neural stimulation: (**a**) Schematic of stimulation and recording process; (**b**) The recorded EMG signals according to the nerve stimulation through DCNI; (**c**) Peak-to-peak amplitude of the EMG signals (n = 60); (**d**) The recorded ENG signals through DCNI according to the nerve stimulation through a hook electrode; and (**e**) Peak-to-peak amplitude of the CNAP (n = 60).

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
