# Peer review of "Advanced Neural Interface toward Bioelectronic Medicine Enabled by Micro-Patterned Shape Memory Polymer"

_micromachines, 2021, doi:10.3390/mi12060720_

Round 1

Reviewer 1 Report

Really a nice paper with interesting results. However, in the first sections, the text Needs to be double checked for basic wirting such aus "interferes worsens" or "thinker".

In the introduction you state "Silicone and PDMS" These are two different categories of materials.

Does a nerve come with > 2 GPa?

You mention thet the biocompatibility of the SMP has bee proven elsewhere. Please add, how and by which tests and whether they are indicating later chronical use.

Is "clipping" the right word? Do you mean "hugging"

Did you check the thightness of IrO2 on top of the Au layer? Can you report on the long term stability of such layers?

As far as I undestood, you were performing Stimulation with your device. Is an impedance in the range of 3,5 kOhms really sufficient for stimulation? As far as known to me, the electrphysiologists would like to see 100 ohms.

Reviewer 2 Report

This work by Cho et al. describes the development of a “neural interface” electrode based on shape memory polymer. The authors explain that the electrode is implanted into a nerve in a rodent model. However, the work does not provide any clear advantage over the multiple papers on neural interfaces, instead, it seems to repeat what is currently possible with any other materials. Moreover, the electrode is not fully characterized as expected in the field. In general, the authors aimed to provide a simple engineering design of yet another neural interface electrode, although in my opinion, they do not provide proper material and performance characterization. The work is too early to be published, but I would consider it after the electrode and performance are better characterized. I have the following concerns:

  1. Basic scholarly reporting standards such as number of repetitions or error bars are missing. Include them in main text.
  2. The captions of the figures do not provide enough information, I would suggest expanding each figure description.
  3. There are some typos, eg “thinker” page 2 line 54
  4. Provide any type of control or comparison between the proposed hook deformable electrode and a typical neural interface, eg cuff electrode. Or any other experiment that justifies this prototype over the state of the art.
  5. I suggest expanding the electrode characterization under different mechanical stress and bending of the hook.

Round 2

Reviewer 2 Report

The authors expanded some control experiments and corrected some grammar errors, although I don't see a great innovation over the state of the art the article now has a better scholarly presentation thus I approved its publication in Micromachines.